# Towards Age Determination of Southern King Crab (*Lithodes santolla*) Off Southern Chile Using Flexible Mixture Modeling

**Javier E. Contreras-Reyes** [1,*] **, Freddy O. López Quintero** [2] **and Alejandro A. Yáñez** [3]

1 Instituto de Estadística, Universidad de Valparaíso, Valparaíso 2360102, Chile
2 Telefónica I+D, Compañía de Telecomunicaciones de Chile S.A., Santiago 7501105, Chile;
  freddy.lopez.quintero@gmail.com
3 Departamento de Evaluación de Recursos, Instituto de Fomento Pesquero, Valparaíso 2361827, Chile;
  alejandro.yanez@ifop.cl
* Correspondence: jecontrr@uc.cl; Tel.: +56-9-7960-8218

**Abstract:** This study addresses the problem of age determination of the southern king crab (*Lithodes santolla*). Given that recapture is difficult for this species and, thus, age cannot be directly determined with the help of the annual marks on the shell, the von Bertalanffy growth function (vBGF) cannot be used to directly model length-frequency data (LFD). To determine age classes, some researchers have proposed using the MIX algorithm that consists of sampling realization of a finite mixture of normal (FMN) distributions for each LFD. However, normality assumption in age-length data has been questioned in several works related to fish growth analysis. For this study, we considered the biological information of the southern king crab for the period 2007–2015 and localization between $50°06'$–$53°15'$ S and $76°36'$–$72°18'$ W. We assumed that LFD could be modelled by the novel class of finite mixture of skew-*t* (FMST). Assigned age classes were used to estimate the vBGF parameters. The estimated vBGF parameters were $L_\infty = 176.756$ cm, $K = 0.151$ year$^{-1}$, $t_0 = -1.678$ year for males, and $L_\infty = 134.799$ cm, $K = 0.220$ year$^{-1}$, $t_0 = -1.302$ year for females. This study concludes that (a) FMST modal decomposition can detect a group of younger individuals at age 2, given that those individuals have LFD with a left heavy-tail and asymmetry; (b) FMST produces a better representation of LFD than the FMN model; (c) males have bigger $L_\infty$ but grow slower than females; and (d) as expected, a high correlation exists among the vBGF estimates.

**Keywords:** finite mixture; skew-*t* distribution; southern king crab; length-frequency data; von Bertalanffy

## 1. Introduction

Exploitation of the southern king crab started in 1928 off the west coast of Tierra del Fuego (austral Chilean waters) [1]. This fishery is crucial for the local economy as it represents the main fishing activity (80% of all fisheries) in the Magallanes Region [2]. However, catches have also been reported from the city of Valdivia in the Los Ríos region ($50°06'$–$55°59'$ S). The Servicio Nacional de Pesca (Chilean National Fishery Service) reported 6490 tons of landings during 2012, the highest value of landings in southern king crab fishery history [3].

Female southern king crabs reach maturity at 86.51 mm at $L_{50}$ (50% of female population). Vinuesa et al. [4] described that length-frequency data (LFD) aspects, such as molt frequency, decrease with age. Female southern king crabs molt six to seven times in the first year, four to five times in the second, and three times in their third year. From then on, females molt annually and start channeling energy toward gonad development. Estimates of von Bertalanffy [5] growth parameters and age

composition for the southern king crab are a key issue in population dynamic models used for stock assessment [6]. Southern king crabs grow mainly through the increment in size per molt and molting frequency over a period of time. Growth models consider those discontinuities in the growth process. Annual marked tagging is possible for this species; however, recapture proves difficult.

Many researchers found that growth models such as the von Bertalanffy growth function (vBGF) can be used to model LFD, especially for species that do not directly show ageing with annual marks on their shell. An early work by Roa and Tapia [7] considers the MIX algorithm [8], which consists of sampling realization of a mixture of probability distributions for each red squat lobster *Pleuroncodes monodon* LFD and the visual identification of modes. Parameter estimates of each mixture are obtained via maximum likelihood and assuming normally distributed year classes. Given the assigned groups, it is possible to consider a respective age for each mode and, therefore, an age group for each carapace-length distribution. This way, a growth function is fitted to this set of observed  carapace length by unobserved age, being, for example, the von Bertalanffy [5], Richards [9], or Schnute functions [10,11]. MULTIFAN is a more sophisticated method than MIX because the mean of assumed normal distribution is obtained directly from vBGF estimates, creating a more realistic log likelihood from LFD [12]. MULTIFAN also identifies the ages as a nonobservable variable derived from the classification of a set of groups or modes from a typical mixture of normal densities. The number of mixture-components can be detected via the information criteria for model selection, such as Akaike's (AIC) or Bayesian (BIC) information criteria, or given by expert criteria [13].

Roa-Ureta [11] considers a more robust estimation method with multivariate LFD analysis. Beyond that, MULTIFAN does not accomplish regularity conditions for a likelihood ratio test and, because parameters are uncorrelated, Roa-Ureta's method accounts for the complete multivariate structure by considering each vector of mean-length estimates from each LFD set as the unitary observation value in Schnute's growth model, a more general interpretation of red squat lobster LFD. More recently, Yáñez et al. [3] studied the southern king crab *Lithodes santolla* LFD using methods of modal decomposition of LFD described in Reference [14]. Both assumed that each modal component corresponds to one age group, and its identification is based on Finite Mixture of Normal (FMN) densities. In addition, they assumed different initial and fixed $L_\infty$ parameters, and then estimated $K$ and $L_0$ (the first observed carapace length in LFD). The best model was chosen according to AIC criteria.

However, normality assumption in age-length data has also been discussed in several investigations related to fish growth analysis.  For example, Contreras-Reyes et al. [15], López Quintero et al. [16], and Contreras-Reyes et al. [17] considered skewed and heavy-tailed errors in the vBGF through the presence of extreme observations in southern blue whiting (*Micromesistius australis*) and pink cusk eel (*Genypterus blacodes*).  See references therein for more examples of the use of skewed and heavy-tailed distributions in the vBGF model errors. To find the Kaplan–Meier survival estimates and growth in three pelagic larval stages from three populations in the Northwest Atlantic, Ouellet et al. [18] considers the skew-normal (SN) [19] distribution to account for asymmetry in survival data, where they detect higher values toward higher temperatures.

Typical distribution of carapace length for each cohort at recruitment is assumed normal by researchers, so the mixture models for each LFD are considered normal mixture-distributed, where each age component is an annual cohort [7,11].  In this paper, we consider the proposed methodology [11] assuming that LFD could be modeled by the novel class of a finite mixture of skew-*t* (FMST) [20]. Both classes provide some advantages over the FMN model. For instance, normal components can lead to wrong classification because they allow an arbitrarily close modeling of any distribution by increasing the number of groups of carapace lengths represented by asymmetrically distributed LFD [20]. The FMST components capture both skewness and extreme carapace lengths due to their flexibility. Thus, this work considers the scale mixtures of SN (SMSN) family distributions oriented to finite mixtures, which includes as a particular case the FMST model. This model is used first to determine modes of southern king crab LFD on each year sample, and for individuals classified by sex (males and females).  In a second stage, age determination using Roa-Ureta's procedure is

considered based on previously obtained modes. In a third stage, vBGF is used for the age-length modeling determined in the second stage.

## 2. Methods

A finite-mixture model is a combination of two or more probability density functions (pdf), allowing to approximate any arbitrary distribution with the mixture of the most used normal ones [21]. This capability has been used in several applications in fishery science, as well as catch-rate standardization [21] and age-length analysis [22]. For the latter, the number of year classes composing each mixture of normal distribution is unknown given that age is a latent variable. This issue is treated as a model identification problem, where selection criteria are commonly used for this effect [11].

Regarding model selection, AIC and BIC criteria are used for selection in age-length models [11,15,22]. BIC presents some advantages over AIC criteria: (a) BIC is more appropriate for large samples because it penalizes the log-likelihood function using sample size and number of parameters, and (b) AIC is less penalized by the number of parameters (parsimonious models) than BIC [15].

Below, we propose a mixture of flexible models to describe the LFD and determine the age of southern king crabs (*Lithodes santolla*) off Chile.

### 2.1. Finite Mixtures of Flexible Distributions

The pdf of an *m*-component mixture model with parameter vector set $\boldsymbol{\theta}$ is

$$f(y; \boldsymbol{\theta}, \text{ß}) = \sum_{i=1}^{m} \pi_i \, f(y; \boldsymbol{\theta}_i), \tag{1}$$

where $\text{ß} = (\pi_1, \ldots, \pi_m)$ is a vector of mixing weights $\pi_i$, with $\pi_i \geq 0$, $\sum_{i=1}^{m} \pi_i = 1$, and $f(y; \boldsymbol{\theta}_i)$ a particular pdf depending on a flexibility degree that could be selected. In a general context, Basso et al. [23] considered SMSN distributions as a robust estimation method of finite components. Random variable $Y$ follows the SMSN family if it can be written as

$$Y \stackrel{d}{=} \mu + \kappa^{1/2}(U)Z, \tag{2}$$

where $\mu$ is a location parameter, $\kappa(\cdot)$ a positive weight function, $U$ a random variable with distribution function $H(\cdot; \nu)$ and density $h(\cdot; \nu)$, $\nu$ is a scalar or vector parameter indexing the distribution of $U$ and $Z \sim SN(0, \sigma^2, \eta)$. From (2), the pdf of $Y$ conditional on $U = u$ is $Y|U = u \sim SN(\mu, \kappa(u)\sigma^2, \eta)$ and the density of $Y$, denoted as $Y \sim SMSN(\mu, \sigma^2, \eta, H)$, is

$$f(y) = 2 \int_0^\infty \phi(y; \mu, \kappa(u)\sigma^2) \Phi[\kappa(u)^{-1/2}\eta\sigma^{-1}(y - \mu)] dH(u; \nu). \tag{3}$$

The generalization of skewed distributions (3) has been used in age-length modeling by Contreras-Reyes et al. [15] and López Quintero et al. [16], mainly through the following:

(a) when $U \sim Gamma(\nu/2, \nu/2)$ in (3), $\nu > 0$, we obtain the skew-*t* distribution (ST) [24] denoted by $Z \sim ST(\xi, \sigma^2, \eta, \nu)$ and with the pdf given by

$$f(z) = \frac{2}{\sigma} t(z_0; \nu) T\left(\eta z_0 \sqrt{\frac{\nu + 1}{\nu + z_0^2}}; \nu + 1\right), \tag{4}$$

where $z_0 = (z - \xi)/\sigma$, $t(z_0; \nu)$ and $T(z_0; \nu)$ denote the pdf and cumulative density function (cdf) of the standard *t* distribution with $\nu$ degrees of freedom, respectively. ST distribution was introduced to achieve a higher degree of excess kurtosis produced by extreme observations. ST distribution converges to SN distribution as $\nu \to \infty$ and is the *t* distribution when $\eta = 0$;

(b)   The stochastic representation (2) of SMSN allows to determine the exact density of conditional distributions necessary for the ECME algorithm. That is, using Lemma 2 of Basso et al. [23], variable $Y$ in (2) is represented conveniently by a hierarchical representation and, in this form, makes it possible to obtain the conditional maximization (CM) steps.

Considering the finite mixture of distributions (1) of SMSN distributions (FM-SMSN), we can determine the FM-SMSN model by using density $f(y)$ for $f(y; \theta_i)$, with $\theta_i = (\xi_i, \sigma_i^2, \eta_i, \nu)$, $i = 1, \ldots, m$, concerning the parameter $\nu$ of the mixing distribution $H(\cdot; \nu)$, taking into account that $\nu$ is assumed for all components $i = 1, \ldots, m$ for computational convenience. Following Shertzer et al. [25], to capture the group membership of southern king crab $i$, we considered latent indicator variable $Z_i$ such that $P(Z_i = 1) = 1 - P(Z_i = 0) = \pi_i$, $i = 1, \ldots, m$, $\sum_{i=1}^{m} Z_i = 1$ and $Y|Z_i = 1 \sim SMSN(\theta_i)$. It can be noted that $Z = (Z_1, \ldots, Z_m)$ follows a multinomial distribution with pdf

$$f(z) = \pi_1^{z_1} \pi_2^{z_2} \cdots (1 - \sum_{i=1}^{m-1} \pi_i^{z_i})^{z_m},$$

which is denoted by $Z \sim M(1; \pi_1, \ldots, \pi_m)$. As is described in (c), $Z$ appears in the hierarchical representation using Lemma 2 of Basso et al. [23] for parameter estimation via the ECME algorithm.

In this paper, we considered the FMST distribution as a robust method for components estimation and its log-likelihood function that generalized the information proportioned by FMN and FMSN distributions. In (1), we considered the parameter vector set $\theta = (\xi, \Sigma, \eta, \nu)$, where $\xi$, $\Sigma$ and $\eta$ are defined as in a); $\nu = (\nu_1, \ldots, \nu_m)$ is a set of $m$ degree of freedom parameters, and $f(y; \theta_i)$ is defined as in (4) with $\theta_i = (\xi_i, \sigma_i^2, \eta_i, \nu)$, $i = 1, \ldots, m$.

## 2.2. Growth Modeling

The vBGF model [5] explains the carapace length of one individual in terms of its age by means of nonlinear function

$$L(x) = L_\infty (1 - e^{-K(x - t_0)}), \tag{5}$$

which also depends on three parameters: $L_\infty$ (cm), the asymptotic carapace length of the species; $K$ (year$^{-1}$), the growth rate coefficient; and $t_0$ (year)—age at zero carapace length. To fit Model (5) from an empirical dataset, $(y_i, x_i)$, $i = 1, ..., n$, the vBGF model can be described in terms of an additive nonlinear regression, $y_i = L_i + \varepsilon_i$, where $y_i$ is the $i$th observed carapace length at age $x_i$, $L_i = L(x_i)$, and $\varepsilon_i$ are independent and identically distributed (iid) $N(0, \sigma^2)$ random errors [26].

The vBGF parameters are estimated from an observed age-length pair $(x_i, L(x_i))$, $i = 1, \ldots, n$, where $L(x_i)$ is the $i$th observed carapace length at age $x_i$.

## 2.3. Implementation

Methods described in Sections 2.1 and 2.2 are implemented as follows:

1.  **Modal decomposition.** The FMST model was carried out for southern king crab LFD with $\nu$ degrees of freedom and $m$ components by zone, year, and sex. For example, degrees of freedom $\nu = 5$ indicate a high presence of heavy-tails in LFD [15,16]. The order of $m$ depends on the reported BIC for each combination, where the 'best' model for each $m$ is selected through the smallest BIC.

2.  **Age-class assignment.** From Step 1, take into account that $m$ modes provide $m$ modes used for age-class determination, which is at least $m$. Considering the classified carapace lengths, a bar chart of means is built where gray and white colors alternate and represent classified age classes. The cluster means are ordered and grouped into cohorts, so that no year is repeated within each group. "Premise II" of Reference [11] is considered as a criterion to determine the cohort point between age classes; textually, this is: *strong assumption that each year no more than one cohort* (means that it is not possible that two mean carapace lengths with the same year index fall into the same

age class) *and no less than one cohort* (means that it is not possible that two consecutive mean carapace lengths with the same year index fall into two nonconsecutive age classes) *enters the population.* Given that age classes 0 and 1 include individuals with molt frequency decreasing with age (six to seven molts in the first and four to five in the second year), we opted to label the first group with Year '2' and then estimate the vBGF parameters.

3.  **vBGF model.** Given the estimated year in Step 2, the formed age-length data serve to evaluate the vBGF (5) growth function.

## 3. Computational Implementation

*3.1. Data*

The data analyzed in this study correspond to biological information of the southern king crab for the 2007–2015 period and localization 50°06′–53°15′ S and 76°36′–72°18′ W. Observations localized between 54°13′–55°59′ S and 69°40′–66°41′ W for the 2007–2010 period were not considered because they present too few individuals and years (see Table 1). Figure 1 shows the spatial distribution of both groups. Northern individuals are concentrated off the city of Puerto Natales, while southern individuals are concentrated off the town of Cabo de Hornos (on the border with Ushuaia, Argentina). The individuals were differentiated by sex (males and females), and, for each individual, carapace lengths from 14 mm (males) to 212 mm (females) were considered.

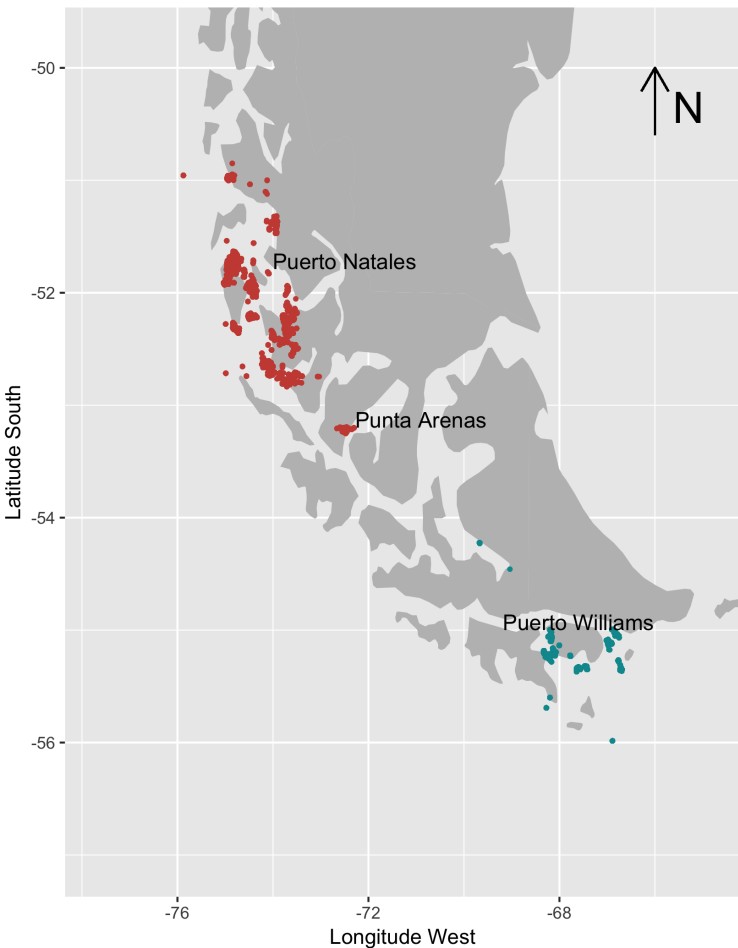

**Figure 1.** Study area restricted to Chile (50°06′–55°59′ S, 76°36′–66°41′ W). Red and green points correspond to northern (50°06′–53°15′S, 76°36′–72°18′ W) and southern (54°13′–55°59′ S, 69°40′–66°41′ W) individuals, respectively.

**Table 1.** Descriptive statistics for the southern king crab grouped by sex and year. LQ and UQ stand for Lower Quantile and Upper Quantile, respectively.

| Sex | Year | Min. | LQ | Median | Mean | UQ | Max. | SD | *n* |
|-----|------|------|-----|--------|------|-----|------|------|-----|
| Males | 2007 | 34 | 111 | 122 | 120.1 | 132 | 176 | 17.86 | 1734 |
| | 2008 | 39 | 112 | 122 | 121.340 | 132 | 193 | 18.567 | 4983 |
| | 2009 | 31 | 113 | 125 | 123.604 | 136 | 177 | 18.158 | 5759 |
| | 2010 | 39 | 104 | 116 | 115.230 | 127 | 171 | 18.261 | 3424 |
| | 2011 | 14 | 106 | 121 | 118.791 | 134 | 175 | 21.725 | 6282 |
| | 2012 | 60 | 108 | 119 | 116.589 | 128 | 170 | 16.636 | 5453 |
| | 2013 | 34 | 113 | 124 | 121.719 | 134 | 179 | 18.437 | 4256 |
| | 2014 | 39 | 115 | 126 | 125.210 | 136 | 182 | 17.499 | 80,612 |
| | 2015 | 58 | 107 | 116 | 115.510 | 126 | 171 | 16.132 | 13,683 |
| Females | 2007 | 35 | 100 | 112 | 110.730 | 122 | 162 | 16.426 | 1586 |
| | 2008 | 35 | 104 | 113 | 112.751 | 122 | 165 | 15.365 | 5883 |
| | 2009 | 33 | 105 | 113 | 113.319 | 121 | 168 | 14.105 | 7717 |
| | 2010 | 46 | 98 | 108 | 107.192 | 116 | 165 | 15.137 | 3273 |
| | 2011 | 30 | 94 | 105 | 105.844 | 117 | 165 | 17.602 | 7366 |
| | 2012 | 61 | 93 | 101 | 100.306 | 108 | 152 | 11.554 | 5530 |
| | 2013 | 43 | 102 | 110 | 109.949 | 118 | 167 | 14.113 | 3508 |
| | 2014 | 38 | 104 | 112 | 111.552 | 120 | 212 | 14.364 | 59,090 |
| | 2015 | 58 | 97 | 104 | 103.903 | 112 | 144 | 11.156 | 22,852 |

*3.2. Computational Aspects*

All models and summaries in this article were estimated with free software environment for statistical computing and graphics R [27]. All computational estimations was made under Linux v. 4.15 and MacOS v. 10.13 operating systems. Particularly, to estimate the mixture of distributions, we used the mixsmsn package, developed by Prates et al. [28]. The mixsmsn package considers the Expectation-Maximization algorithm [29] for FMST modal decomposition. For the vBGF estimates and initial values, the nls and FSA packages were used, respectively. More details appear in Reference [30].

## 4. Results

Table 1 provides descriptive statistics by sex and year. Included are minimum, lower quantile, median, mode, mean, upper quantile, maximum, and standard deviations. Our main interest was on the growth-pattern differences between males and females for northern individuals, as explained in Section 3.1. At first sight, males are larger than females considering the range of their carapace lengths.

In this section, the methodology described in Section 2 is applied to southern king crab LFD by year and sex. As each step is done for each combination, some verbose repetition is anticipated in the subsections. In addition, FMST models were ran with $m = 2, \ldots, 9$ (the most simple case of $m = 1$ was omitted) and also the respective BIC for each combination.

*4.1. Males*

BIC criteria for the FMST models provided two means for 2007; three means for 2010, 2012, and 2013; five means for 2008, 2009, and 2011; six means for 2014, and eight means for 2015 (Table 2). The BIC values of FMST were also smaller than FMN ones for all years. We could observe that the smallest and largest means were 58.301 and 152.129 mm for 2009 and 2014, respectively (Table 3). In 2009, the smallest mean was provided, which produces a left heavy-tail (Figure 2). This mean represents a mode of male juvenile southern king crab from the northern group that would be helpful in the assignment of the first age class. In general, the FMST model provides good fits of annual LFD and well-separated cohort groups (see panels of Figure 2). Results of 10 assigned age classes are provided in the bar chart of Figure 3a.

**Table 2.** Bayesian information criterion (BIC) values of the finite mixture of normal (FMN) and finite mixture of skew (FMST) fitted models ($m = 2, \ldots, 9$) for each specification (sex and year). The smallest values for each model and year are marked in bold.

| Sex | Model | Year | Number of Modes ($m$) | | | | | | | |
|---|---|---|---|---|---|---|---|---|---|---|
| | | | **2** | **3** | **4** | **5** | **6** | **7** | **8** | **9** |
| Males | FMN | 2007 | **21,660.32** | 25,110.58 | 28,584.18 | 32,027.82 | 35,511.00 | 38,942.26 | 42,404.44 | 45,867.89 |
| | | 2008 | **62,812.42** | 72,732.02 | 82,687.67 | 92,642.98 | 102,607.25 | 112,582.18 | 122,534.11 | 132,498.96 |
| | | 2009 | **72,131.15** | 83,620.96 | 95,128.46 | 106,629.39 | 118,143.52 | 129,656.02 | 141,144.20 | 152,665.18 |
| | | 2010 | **43,261.16** | 50,089.41 | 56,927.93 | 63,774.23 | 70,585.84 | 77,434.39 | 84,261.89 | 91,108.33 |
| | | 2011 | **81,316.49** | 93,847.13 | 106,403.04 | 118,966.49 | 131,524.34 | 144,081.13 | 156,638.06 | 169,188.31 |
| | | 2012 | **67,406.53** | 78,299.57 | 89,191.86 | 100,106.07 | 111,008.67 | 121,889.19 | 132,817.15 | 143,720.88 |
| | | 2013 | **53,475.91** | 61,958.09 | 70,463.77 | 78,974.94 | 87,477.20 | 95,984.53 | 104,494.43 | 113,008.14 |
| | | 2014 | **1,010,474.45** | 1,171,030.24 | 1,331,999.31 | 1,493,074.55 | 1,654,157.23 | 1,815,515.11 | 1,976,135.29 | 2,137,650.73 |
| | | 2015 | **169,223.51** | 196,429.18 | 223,791.31 | 251,162.38 | 278,521.06 | 305,856.42 | 332,855.51 | 360,116.56 |
| | FMST | 2007 | **13,364.51** | 13,371.75 | 13,375.64 | 14,843.43 | 14,878.02 | 14,892.64 | 14,910.46 | 14,932.68 |
| | | 2008 | 48,624.99 | 48,585.88 | 48,595.47 | **42,986.84** | 43,009.54 | 43,059.00 | 43,061.88 | 43,096.80 |
| | | 2009 | 62,001.87 | 61,869.83 | 61,871.81 | **49,233.68** | 49,267.38 | 49,274.14 | 49,321.24 | 49,333.27 |
| | | 2010 | 27,032.18 | **27,020.63** | 27,026.80 | 29,699.00 | 29,729.53 | 29,749.19 | 29,785.83 | 29,779.53 |
| | | 2011 | 63,056.19 | 63,046.12 | 62,964.81 | **56,325.63** | 56,358.11 | 56,391.05 | 56,417.41 | 56,439.62 |
| | | 2012 | 42,731.45 | **42,730.32** | 42,738.57 | 45,750.64 | 45,783.98 | 45,817.96 | 45,845.73 | 45,874.98 |
| | | 2013 | 28,366.28 | **28,334.04** | 28,340.50 | 36,570.12 | 36,607.13 | 36,640.77 | 36,676.77 | 36,698.06 |
| | | 2014 | 687,830.5 | 687,524.6 | 687,307.9 | 687,305.38 | **687,289.56** | 687,347.91 | 687,457.90 | 687,328.14 |
| | | 2015 | 114,496.6 | 114,461.6 | 114,494.8 | 114,453.80 | 114,554.55 | 114,353.79 | **114,252.30** | 11,4437.46 |
| Females | FMN | 2007 | **19,713.96** | 22,852.88 | 26,023.90 | 29,192.11 | 32,359.81 | 35,532.03 | 38,696.17 | 41,875.65 |
| | | 2008 | **72,139.24** | 83,843.33 | 95,605.08 | 107,358.69 | 119,124.74 | 130,859.57 | 142,656.47 | 154,376.82 |
| | | 2009 | **92,895.77** | 108,137.59 | 123,550.34 | 138,974.63 | 154,408.95 | 169,848.59 | 185,267.58 | 200,701.04 |
| | | 2010 | **40,113.74** | 46,619.98 | 53,153.97 | 59,699.58 | 66,235.46 | 72,768.60 | 79,308.71 | 85,864.49 |
| | | 2011 | **92,517.32** | 107,138.55 | 121,826.14 | 136,551.47 | 151,272.19 | 166,021.39 | 180,718.50 | 195,463.84 |
| | | 2012 | **64,868.23** | 75,887.82 | 86,941.12 | 97,998.73 | 109,057.37 | 120,093.74 | 131,144.66 | 142,200.95 |
| | | 2013 | **42,350.73** | 49,353.13 | 56,369.57 | 63,378.44 | 70,397.06 | 77,409.93 | 844,25.82 | 91,438.30 |
| | | 2014 | **716,320.29** | 834,395.94 | 951,934.44 | 1,070,071.78 | 1,188,213.60 | 1,306,378.32 | 1,424,541.82 | 1,542,624.05 |
| | | 2015 | **266,204.82** | 311,746.05 | 357,317.35 | 403,020.59 | 448,584.81 | 494,297.28 | 539,896.96 | 585,635.92 |
| | FMST | 2007 | **13,402.09** | 13,430.81 | 13,456.18 | 13,478.17 | 13,504.66 | 13,532.48 | 13,553.48 | 13,570.40 |
| | | 2008 | 48,671.75 | **48,659.36** | 48,695.67 | 48,725.50 | 48,763.35 | 48,781.16 | 48,803.79 | 48,834.31 |
| | | 2009 | 62,050.53 | 61,946.29 | 61,976.07 | 62,004.46 | 62,041.64 | 62,066.87 | 62,104.48 | **61,535.19** |
| | | 2010 | 27,074.84 | **27,087.65** | 27,118.21 | 27,143.20 | 27,167.77 | 27,189.12 | 27,213.60 | 27,237.89 |
| | | 2011 | 63,104.52 | 63,122.07 | **63,068.38** | 63,074.54 | 63,102.86 | 63,149.36 | 63,181.02 | 63,210.95 |
| | | 2012 | **42,777.77** | 42,803.12 | 42,837.84 | 42,876.89 | 42,906.37 | 42,925.71 | 42,971.24 | 42,985.32 |
| | | 2013 | 28,409.42 | **28,401.83** | 28,432.94 | 28,465.27 | 28,492.60 | 28,527.44 | 28,556.07 | 28,583.88 |
| | | 2014 | 480,353.4 | 480,016.9 | 480,040.3 | 479,967.99 | 479,778.60 | 479,668.53 | **479,583.39** | 479,593.39 |
| | | 2015 | 174,927.2 | 174,939.8 | 174,752.3 | 174,863.17 | 174,746.23 | 174,709.63 | 174,691.60 | **174,687.12** |

**Table 3.** Estimate modes for the southern king crab using the FMST model for each sex and year.

| Sex | Year | $\mu_1$ | $\mu_2$ | $\mu_3$ | $\mu_4$ | $\mu_5$ | $\mu_6$ | $\mu_7$ | $\mu_8$ | $\mu_9$ |
|-----|------|------|------|------|------|------|------|------|------|------|
| Males | 2007 | 117.098 | 120.580 | - | - | - | - | - | - | - |
| | 2008 | 79.255 | 103.574 | 121.668 | 126.899 | 142.613 | - | - | - | - |
| | 2009 | 58.301 | 102.911 | 115.216 | 122.973 | 138.269 | - | - | - | - |
| | 2010 | 101.505 | 119.793 | 125.396 | - | - | - | - | - | - |
| | 2011 | 81.542 | 100.494 | 116.096 | 123.987 | 140.691 | - | - | - | - |
| | 2012 | 91.457 | 121.495 | 125.046 | - | - | - | - | - | - |
| | 2013 | 94.073 | 126.019 | 128.364 | - | - | - | - | - | - |
| | 2014 | 95.159 | 115.189 | 126.915 | 131.681 | 142.104 | 152.129 | - | - | - |
| | 2015 | 76.686 | 82.263 | 98.657 | 112.971 | 119.165 | 130.680 | 141.562 | 149.571 | - |
| Females | 2007 | 106.905 | 115.245 | - | - | - | - | - | - | - |
| | 2008 | 103.470 | 116.723 | 123.758 | - | - | - | - | - | - |
| | 2009 | 60.367 | 63.205 | 81.713 | 93.794 | 100.643 | 107.161 | 111.463 | 121.416 | 132.965 |
| | 2010 | 98.687 | 106.437 | 125.761 | - | - | - | - | - | - |
| | 2011 | 83.598 | 101.347 | 110.181 | 127.319 | - | - | - | - | - |
| | 2012 | 95.546 | 99.692 | - | - | - | - | - | - | - |
| | 2013 | 104.431 | 106.143 | 121.669 | - | - | - | - | - | - |
| | 2014 | 83.145 | 94.126 | 101.085 | 107.903 | 113.760 | 116.014 | 125.594 | 137.902 | - |
| | 2015 | 75.346 | 86.977 | 97.093 | 103.001 | 107.625 | 109.844 | 112.652 | 115.984 | 119.786 |

Assigned age classes are posteriorly used as age-length data for vBGF modeling (Table 4, Figure 3b). We observed that all vBGF parameters were significant and correlated with $L_\infty$, and $K$ is the highest among the three vBGF parameters. For all cases, high and negative correlations between $L_\infty$ and $K$, and $L_\infty$ and $t_0$, respectively, and high and positive correlation between $K$ and $t_0$. We can see that $t_0$ is negative and close to $-1.7$, becoming relevant for younger individuals, as illustrated in Figure 3b. The distance of modes related to the third is more pronounced than in the fourth age class, except for the largest mean (81.542 mm; see also Figure 3a). The means of the oldest, the 10th, age class are close to the previous, the ninth, age class. In general, vBGF fitted in the middle of all modes in each age class (Figure 3b), as observed in the residuals plot of Figure 3c. Generally, residual values are concentrated at approximately 3 mm and indicate a constant trend, thereby suggesting uncorrelated observations.

**Table 4.** Von Bertalanffy growth function (vBGF) estimates, standard error (SE), Student ($t$) value, and $p$-value, $\Pr(> |t|)$, for the southern king crab for each sex. Estimated correlations between vBGF parameters are in the last three columns.

| Sex | Parameter | Estimate | SE | $t$ | $\Pr(> |t|)$ | $L_\infty$ | $K$ | $t_0$ |
|-----|-----------|----------|-----|-----|--------------|-----------|-----|-------|
| Males | $L_\infty$ | 176.756 | 12.865 | 13.739 | <0.001 | 1.000 | - | - |
| | $K$ | 0.151 | 0.032 | 4.773 | <0.001 | $-0.985$ | 1.000 | - |
| | $t_0$ | $-1.678$ | 0.519 | $-3.231$ | 0.002 | $-0.897$ | 0.956 | 1.000 |
| Females | $L_\infty$ | 134.799 | 4.065 | 33.162 | <0.001 | 1.000 | - | - |
| | $K$ | 0.220 | 0.031 | 7.029 | <0.001 | $-0.950$ | 1.000 | - |
| | $t_0$ | $-1.302$ | 0.442 | $-2.946$ | 0.005 | $-0.830$ | 0.952 | 1.000 |

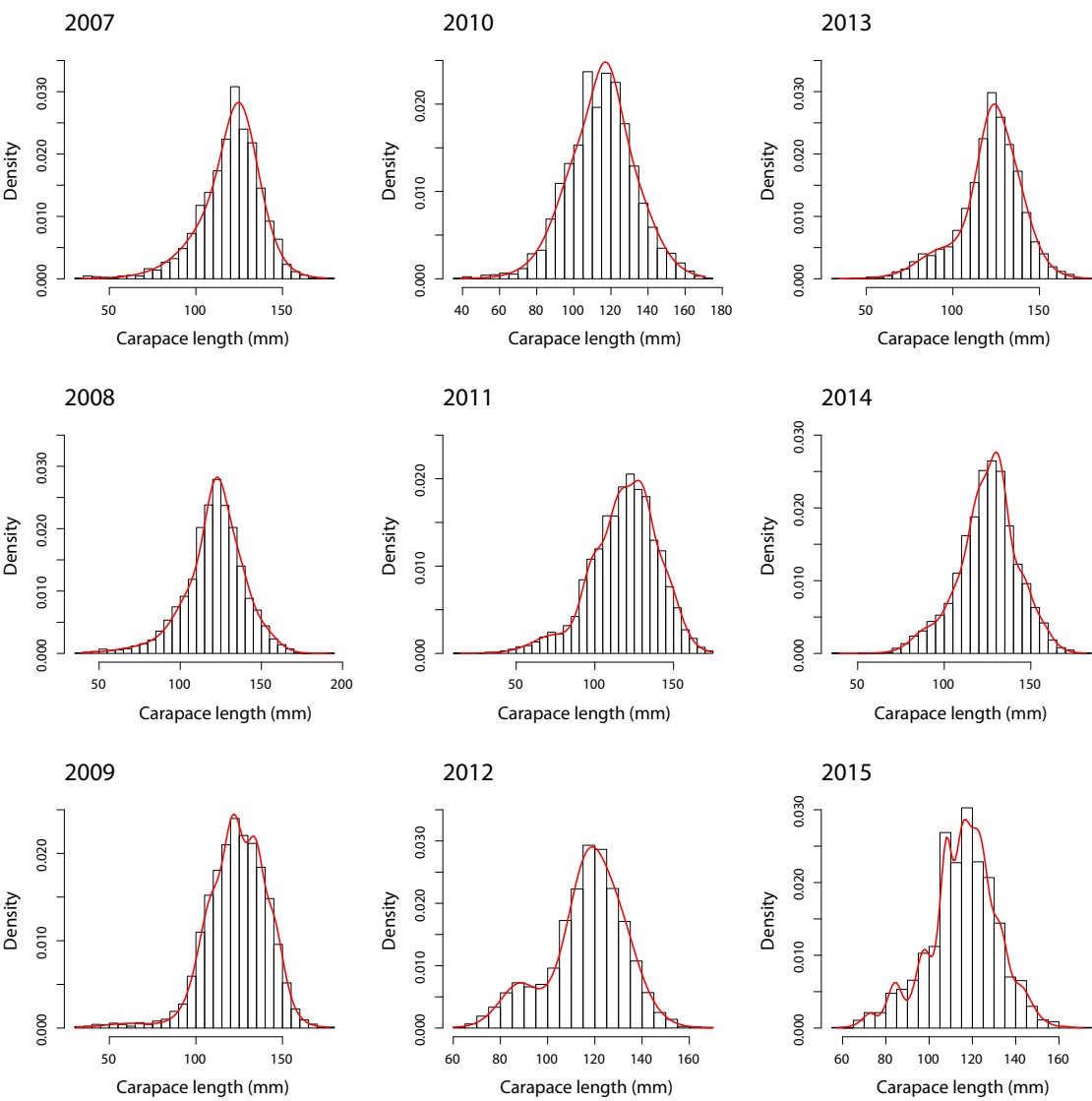

**Figure 2.** FMST fits for males—northern length-frequency data (LFD) datasets and years 2007–2015.

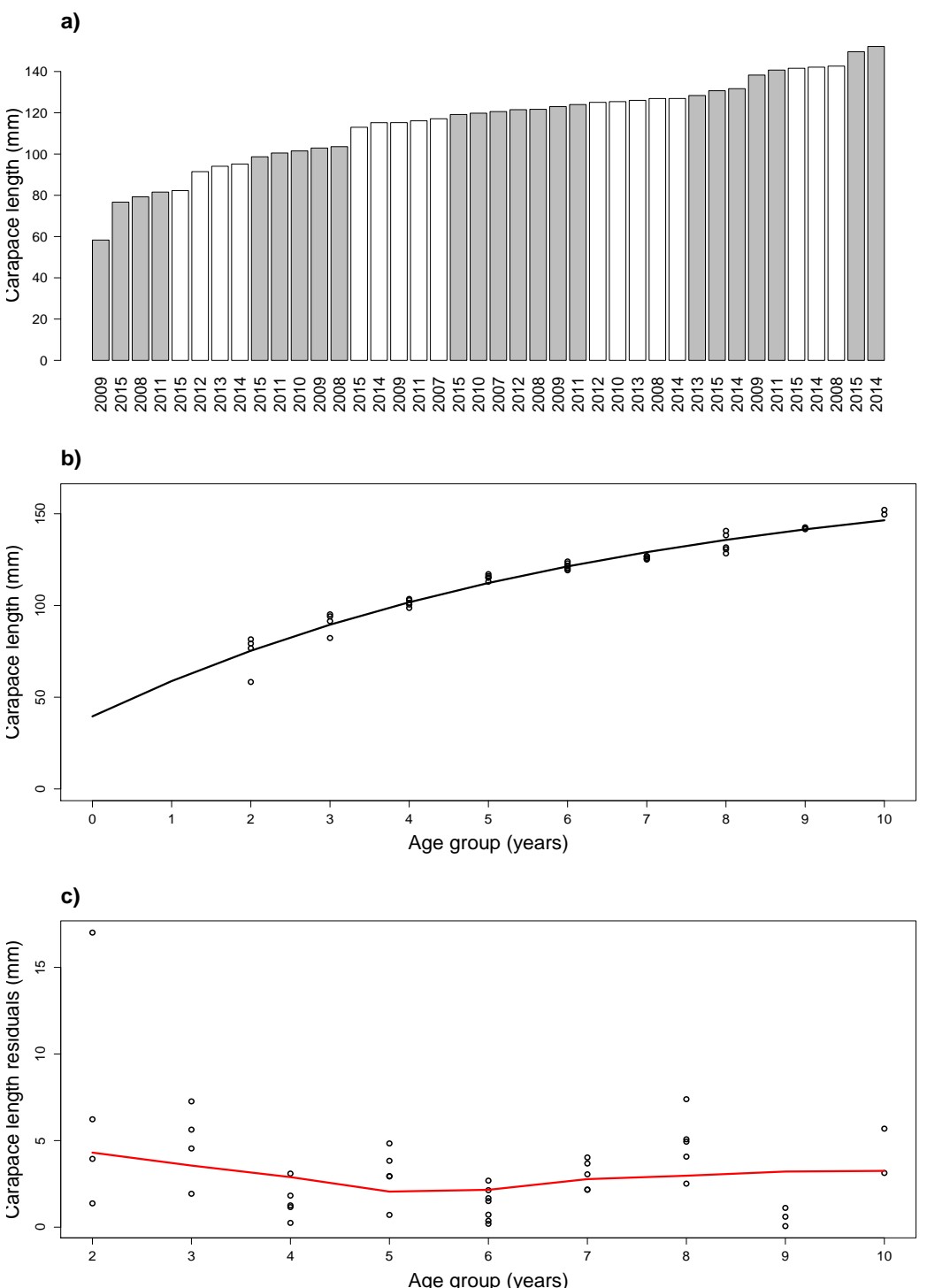

**Figure 3.** (**a**) Barplot of assigned age from FMST modal decomposition, (**b**) vBGF model fits for estimated age-length, and (**c**) its respective absolute residuals for male southern king crabs.

## *4.2. Females*

In Table 3, we observed that the FMST model provides two means for 2007 and 2012, three means for 2008, 2010, and 2013, four means for 2011, eight for 2014, and nine means for 2015; all obtained considering BIC criteria (Table 2). For males, the BIC values of FMST were smaller than those obtained by the FMN model and for all years. We could observe that the smallest and largest means were 60.367 and 137.902 mm for 2009 and 2014, respectively. In 2009, the smallest mean was provided, which

produces a left heavy-tail (Figure 4). This mean represents a mode of female juvenile southern king crab that help in the assignment of the first age class. In general, the FMST model provides good fits of annual LFD and well-separated cohort groups (see panels of Figure 4). Results of assigned age class are provided in the bar chart in Figure 5a. Therefore, LFD were assigned 11 age classes, where the ninth and 11th age classes (oldest individuals) had the smallest number of modes. This information is used as age-length data for vBGF modeling in Figure 5b.

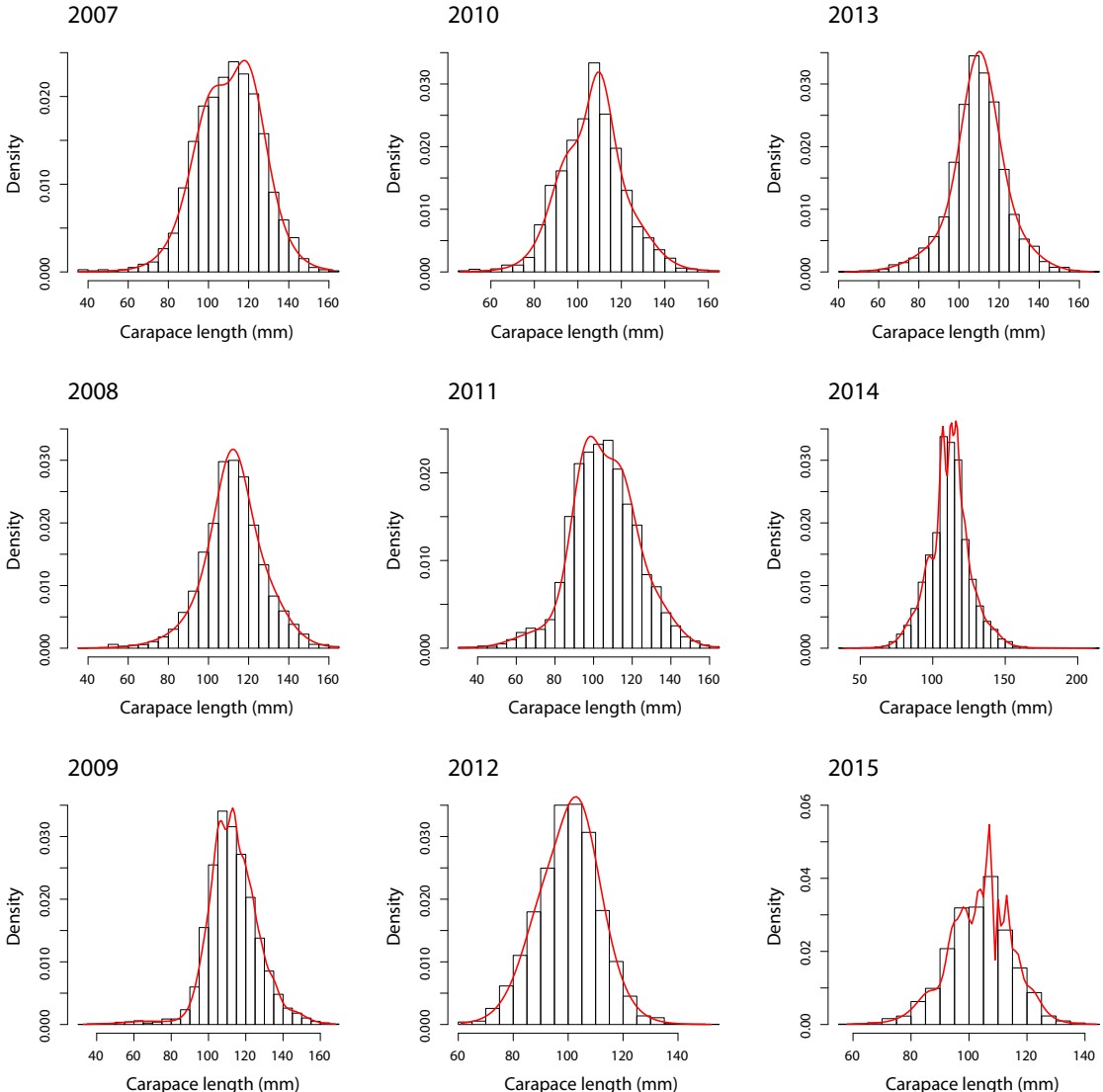

**Figure 4.** FMST fits for Females—northern LFD datasets and years 2007–2015.

vBGF estimates appear in Table 4 and are all significant. For males, high and negative correlations between $L_\infty$ and $K$, and $L_\infty$ and $t_0$, and high and positive correlation between $K$ and $t_0$. We can see that $t_0$ is negative and close to $-1$, taking relevance for younger female individuals (Figure 5b). The vBGF model fit is illustrated in Figure 5b where a clear distance of modes is observed for the third with respect to the fourth age class. The mode of the oldest, the 11th, age class is significantly distant from the previous, the 10th, age class. In contrast to the vBGF fit for males, the vBGF in females was not fitted in the middle of all modes in each age class (seventh–ninth and 11th). However, as can be seen in the residuals plot in Figure 5c, residual values are concentrated at approximately 3 mm and generally indicate a constant trend and uncorrelated observations, except for younger individuals (second age class) and older individuals (7–11th age classes).

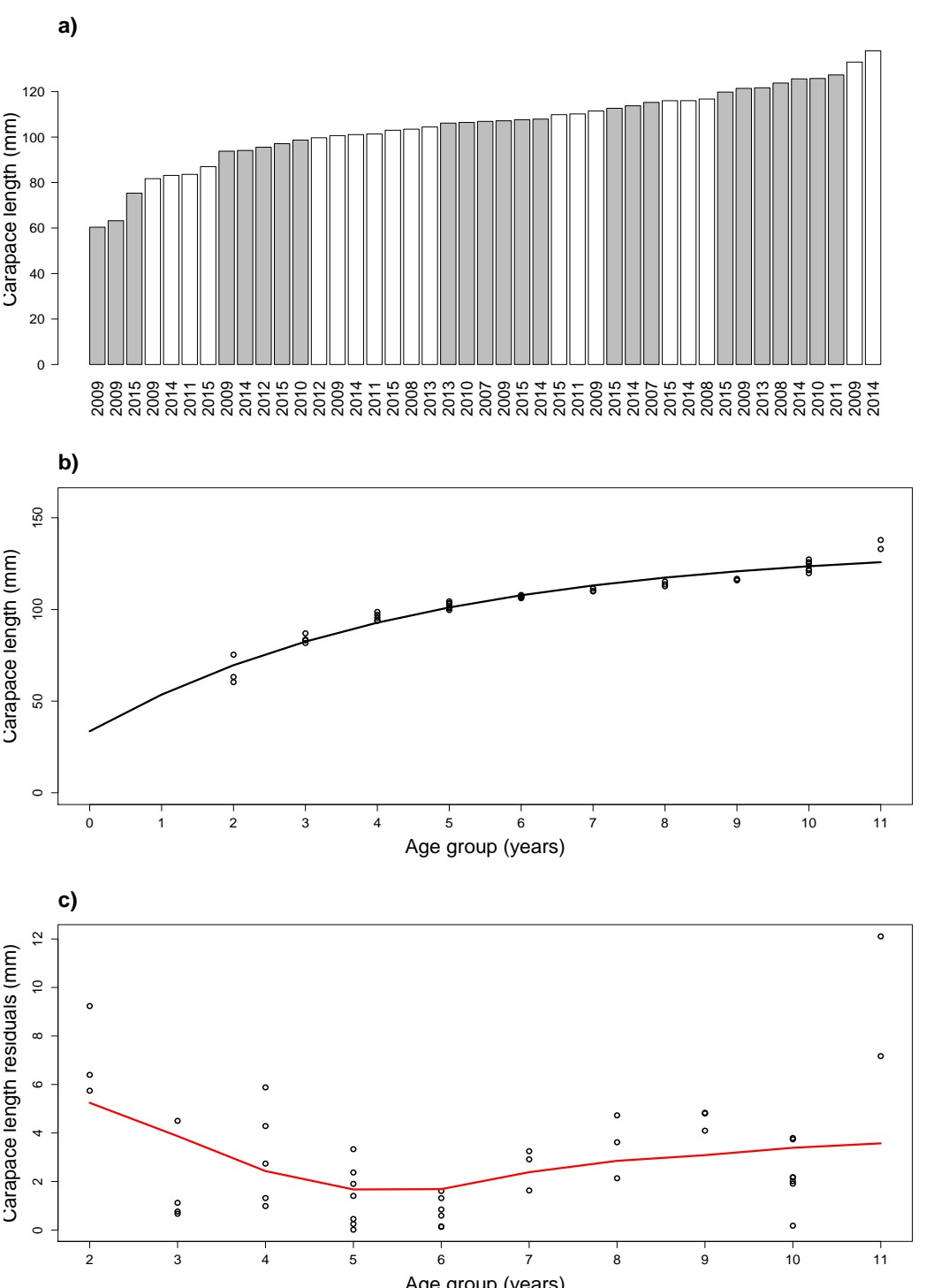

**Figure 5.** (**a**) Barplot of assigned age from FMST modal decomposition, (**b**) vBGF model fits for estimated age-length, and (**c**) its respective absolute residuals for female southern king crabs.

## 5. Discussion

In this study, we addressed the age determination and flexible mixture modeling for the southern king crab off southern Chile. This study mainly suggests that (a) FMST modal decomposition can detect a group of younger individuals at age 2, given that those individuals have LFD with a left heavy-tail and asymmetry; (b) based on BIC values, FMST produces a better representation of LFD

than the FMN model, thus inducing more realistic vBGF estimates; (c) males are larger (biggest $L_\infty$) but grow slower than females; and (d) as expected, high correlation exists among the vBGF estimates.

High molting frequency found in age group 2 explains the better fit of the FMST model, while the FMN model produces a worse fit to the LFD. In the southern king crab, females concentrate their energy on the reproduction process instead of somatic growth (evolutionary strategy), which explains why they grow faster than males trying to reach maturity sooner. Egg production is limited by body size and, therefore, it is an advantage for females grow faster, reach sexually maturity/capacity, and improve survival and competition. In addition, there is often considerable variation in molt increment based on foraging success prior to molt, and the resulting energetic state at the time of molt. Indeed, results assume that differences in size reflect different cohorts. However, it is entirely possible that some of these differences reflect age-independent size variation due to differences in molt increment. High vBGF parameter correlations, especially the negative one between $L_\infty$ and $K$ parameters, are in line with somatic growth and the reproductive process mentioned above.

Yáñez et al. [3] (see Table 13, p. 25 of the Appendix) presented up to 15 age classes for males and females in 1984–1987, for 1996, and in 2007–2014; and for the northern and southern zones (see Section 3.1). Reported vBGF estimates for males were: $\widehat{L}_\infty = 178.09\,[169.82; 186.37]$ cm and $\widehat{K} = 0.14\,[0.12; 0.15]$ (year$^{-1}$); and for females: $\widehat{L}_\infty = 158.95\,[153.43; 164.47]$ cm, $\widehat{K} = 0.16\,[0.15; 0.17]$ (year$^{-1}$). Yañez et al. considered Canales & Arana's approach [14] having the following disadvantages: (i) number of classes is not adequate for each combination from the data used in this work and is arbitrary because LFD changes through the years; and (ii) it does not allow to directly determine $t_0$ estimate, which is found by replacing $\widehat{L}_\infty$ and $\widehat{K}$ in vBGF but standard deviation cannot be determined.

The differences in the estimates of our study and those of Reference [3] can be interpreted in light of our novel incorporation of a wide range of carapace lengths and a larger sample size (Table 1). The approach developed here is more accurate in terms of error description in distribution, as we found a greater presence of extreme values and variability of length-at-age data [15,16]. The comparison of the growth curves by selection criteria generated different growth curves between the sexes of the southern king crab.

For the combinations between the sexes analyzed here, we obtained good fits of vBGF on LFD. However, for females we observed difficulty for vBGF to fit in all assigned ages. Such errors are common; for example, Roa-Ureta [11] had problems with estimation and therefore preferred Schnute's curve instead of a vBGF curve. In some cases, given that the southern king crab grows quickly at first (1–3 years) and then slowly, perhaps a more flexible growth model could be implemented to realize change in age at maturity. Indeed, Ohnishi et al. [31] proposed a variant of vBGF by including two additional parameters: discontinuous change in age at maturity, $t_m$ (year), and growth rate coefficient post-age at maturity (change of growth rate), and $v$ (year$^{-1}$), to define the function

$$T(t) = t - t_0 - \begin{cases} 0, & t < t_m, \\ v(t - t_m), & t \geq t_m, \end{cases}$$

and to be inserted in (5) as $L(t) = L_\infty(1 - e^{-KT(t)})$, with $t = x_i$ (year) as the assigned age. This model represents the time delay to attain a certain body size in $t \geq t_m$ due to change in energy allocation. Consequently, the growth curve becomes biphasic, combining two independent vBGFs. However, inferential aspects must be addressed to biphasic vBGF such as maximum likelihood function and Fisher information matrix derivation.

To determine the start of the age group, a balance was established between computational stability for growth-function fit and LFD modes. We assumed at least two years as the start of the age group because molt frequency is highest in this period [4]. Thereby, vBGF residuals are uncorrelated and with a constant trend among the assigned ages.

The proposed methods for age determination in the southern king crab crucially depend on the available LFD. In the case that all growth stages are not well-represented in the LFDs, results of FMST modal decomposition produce a misspecification of the assigned age with respect to real

age. The criteria used by Reference [11] to determine the cohort point between age classes are very sensible in vBGF estimates. Depending on sample success, it is easy to consider instances where the sample is missing a cohort representative from one or more years. In addition, vBGF estimates are crucial for the study of stock-assessment models [2,3], but are affected by gear selectivity because it produces censored samples when young individuals are missing. Indeed, a lack of comprehensive age information leads to poor understanding of life history schedules, difficulty in the estimation of vBGF parameters necessary for modeling population dynamics and uncertainty. Therefore, the natural path for age-length modeling is direct age determination by growth band counts in the southern king crab [32].

Our proposed method allows one to obtain an estimate of vBGF parameters from a mixture of distributions, but further research about a direct relationship among vBGF estimates and the observed maximum age and carapace length is necessary. Unfortunately, actual relationships are related to fish resources [33].

**Author Contributions:** J.E.C.-R., F.O.L.Q., and A.A.Y. wrote the paper and contributed the reagents/analysis/materials tools; J.E.C.-R. and F.O.L.Q. conceived, designed, and performed the experiments, and analyzed the data. All authors read and approved the final manuscript.

**Funding:** This research received no external funding.

**Acknowledgments:** We are grateful to the Instituto de Fomento Pesquero (IFOP) for providing access to the data used in this work. Special thanks to Erik Daza, Ruth Hernández, Eduardo Almonacid, Rodrigo Wiff, and Mauricio Ibarra for their helpful insights and discussion on an earlier version of this paper. We are sincerely grateful to the two anonymous reviewers for their comments and suggestions that greatly improved the earlier version of this manuscript.

**Conflicts of Interest:** The authors declare that there is no conflict of interest.

## Abbreviations

The following abbreviations are used in this manuscript:

| | |
|---|---|
| AIC | Akaike's information criterion |
| BIC | Bayesian information criterion |
| CM | Conditional maximization |
| ECME | Expectation/conditional maximization either |
| FMN | Finite mixture of normal |
| FMST | Finite mixture of skew-*t* |
| FM-SMSN | Finite mixture of scale mixtures of skew-normal |
| LFD | Length-frequency data |
| LQ | Lower quantile |
| SMSN | Scale mixtures of skew-normal |
| SE | Standard error |
| SN | Skew-normal |
| ST | Skew-*t* |
| UQ | Upper quantile |
| vBGF | von Bertalanffy growth function |

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
