# Peer review of "Towards Age Determination of Southern King Crab (Lithodes santolla) Off Southern Chile Using Flexible Mixture Modeling"

_jmse, doi:10.3390/jmse6040157_

Round 1

Reviewer 1 Report

This study seeks for a method of determining the length at age for southern king crabs.  This is an important problem for crustaceans in general, but in particular for crustaceans that cannot be recaptured to examine the size of individuals longitudinally.  While the approach here seems to be a robust approach, I have several comments about the presentation.

1)      There is a large number of acronyms used in this proposal, and it makes for very difficult reading.  This is particularly problematic for anyone wanting to consider the approach used in this paper, but is not familiar with the various models referenced.

2)      The figures in the paper are not particularly clear for several reasons.  The text is quite small.  They are missing axes labels at times.  The axes labels do not always make a lot of sense (Longitude (mm)?)  What is “longitude”?  Is this carapace length? Carapace width?  The figure captions do a poor job of describing what is being shown – what are the inset graphs in Fig. 3 and 5?

In addition to the above comments on presentation, I also have questions about the methods used that should be addressed in the Discussion.

1)      How are results influenced if the assumption given on line 147 is violated (that there is exactly one cohort per year)?  Depending on sample success, it is easy to consider instances where the sample is missing a cohort representative from one or more years.

2)      There is often considerable variation in molt increment based on the foraging success prior to molt and the resulting energetic state at the time of molt.  The barplots shown in Fig. 3a and 5a assume that differences in size reflect different cohorts.  However, it is entirely possible that some of these differences reflect age-independent size variation due to differences in molt increment.  This should be considered in the Discussion.

Author Response

Dear Reviewer:

We would like acknowledge this careful revision of our manuscript jmse-389757: "Towards age determination of southern king crab (Lithodes santolla) off southern Chile using flexible mixture modeling". We are grateful that this manuscript can be considered for publication after minor revision. We also thank for all your valuable comments and constructive criticism. We have included (see below), a detailed point-by-point response to all your comments and suggestions. The updated lines appear in red in the new manuscript.

This study seeks for a method of determining the length at age for southern king crabs. This is an important problem for crustaceans in general, but in particular for crustaceans that cannot be recaptured to examine the size of individuals longitudinally. While the approach here seems to be a robust approach, I have several comments about the presentation.

There is a large number of acronyms used in this proposal, and it makes for very difficult reading. This is particularly problematic for anyone wanting to consider the approach used in this paper, but is not familiar with the various models referenced.

R: Thanks for your comment. According to JMSE format, we added the Abbreviations" section at page 13 to summarize all these acronyms and avoid confusion. Only MIX and MULTIFAN acronyms have not been added in this section, because they are the original names of these methods and are not used in this work (excerpt in Introduction section).

The Fi gures in the paper are not particularly clear for several reasons. The text is quite small. They are missing axes labels at times. The axes labels do not always make a lot of sense (Longitude (mm)?) What is "longitude"? Is this carapace length? Carapace width? The fi gure captions do a poor job of describing what is being shown - what are the inset graphs in Fig. 3 and 5?

R: Thanks for your observations. Now, the axis of figures have an adequate size, and missing axes of the three panels appear. Captions are fixed too.

In addition to the above comments on presentation, I also have questions about the methods used that should be addressed in the Discussion.

How are results in influenced if the assumption given on line 147 is violated (that there is exactly one cohort per year)? Depending on sample success, it is easy to consider instances where the sample is missing a cohort representative from one or more years.

R: Thank you very much for this observation. At lines 268-276 of Discussion, we added this has possible disadvantage of the proposed method.

There is often considerable variation in molt increment based on the foraging success prior to molt and the resulting energetic state at the time of molt. The barplots shown in Fig. 3a and 5a assume that differences in size reflect different cohorts. However, it is entirely possible that some of these differences reflect age-independent size variation due to differences in molt increment. This should be considered in the Discussion.

R: Thank you very much for this observation. At lines 236-241 of Discussion, we added this observation of the obtained results.

Reviewer 2 Report

Introduction

The introduction is well written and complete.  One suggestion is to add a sentence or two discussing the stock assessment of southern king crab, the growth functions used in that assessment, and how this study could improve or enhance the models used in management.  Also might be useful include some comments about lack of a true aging structure (fish have otoliths, do southern king crab have any aging structures?).  This would provide some rationale for the length-based growth approximations presented in the study.

L 32 What are the magnitude of landings and value?            

Methods

The application of mixture models with skewed distributions is appropriate for the length-frequency datasets available in this study.  Methods are concisely described.  One issues I see is that the authors do not describe how the crabs were captured and they do not address gear selectivity.  Gear selectivity can drastically impact vBGF parameters when small size classes are not present.  This results in a large negative t0 and low K.  One recommendation is to fit another set of vBGF fixing the t0 at 0 (i.e. force the curve through the origin) and add the lines to figures 3 and 5.

L 154 Specify the length type, is this carapace width?

Results

The tables are very informative and figures are clear and provide all the necessary information to support the conclusions.

Fig 2&4 Change x-axis label to appropriate length type

Fig 3&5 Add y-axis label to top figure, change bottom figure y-axis label to ‘carapace width (mm)’,  and label them appropriately as a and b.

Discussion

L229-232 If females devote more energy to reproduction, wouldn’t that leave less for somatic growth and so they would grow slower?  I think what you’re referring to is an evoluationary strategy.  Egg production is limited by body size and so it’s an advantage for females grow faster and reach sexually maturity/capacity, and also to improve survival and competition.  But they reach smaller sizes because of energy allocation.  Males don’t have to worry about egg capacity, so they grow slower but reach larger sizes.  Consider rephrasing these sentence.

Author Response

Dear Reviewer:

We would like acknowledge this careful revision of our manuscript jmse-389757: "Towards age determination of southern king crab (Lithodes santolla) off southern Chile using flexible mixture modeling". We are grateful that this manuscript can be considered for publication after minor revision. We also thank for all your valuable comments and constructive criticism. We have included (see below), a detailed point-by-point response to all your comments and suggestions. The updated lines appear in red in the new manuscript.

Introduction:

The introduction is well written and complete. One suggestion is to add a sentence or two discussing the stock assessment of southern king crab, the growth functions used in that assessment, and how this study could improve or enhance the models used in management. Also might be useful include some comments about lack of a true aging structure ( fish have otoliths, do southern king crab have any aging structures?). This would provide some rationale for the length-based growth approximations presented in the study.

R: Thanks for your observations. We have already mentioned in Introduction section the previous method of growth function estimation (lines 66{70) based on Canales & Arana (2009)'s work. However, we discuss the disadvantages of this method in Discussion section at lines 248-251. On the another hand, we also agree with your observation about the lack of a true aging structure. For this reason, we add a comment in Discussion section (lines 273-276), where direct age determination is necessary for this issue. 

L32: What are the magnitude of landings and value?

R: Thanks for your suggestion. We add these information at lines 34-36.

Methods:

The application of mixture models with skewed distributions is appropriate for the length-frequency datasets available in this study. Methods are concisely described. One issues I see is that the authors do not describe how the crabs were captured and they do not address gear selectivity. Gear selectivity can drastically impact vBGF parameters when small size classes are not present. This results in a large negative t0 and low K. One recommendation is to fit another set of vBGF fi xing the t0 at 0 (i.e. force the curve through the origin) and add the lines to fi gures 3 and 5.

R: Thanks for your comment and observations. We included now a brief description of gear selectivity in captured crabs at lines 271-273. With respect to your recommendation, it is not useful to t another curve by fi xing t0 at 0, because, this is an hypothetical case when all carapace width observations are available, including small size classes. However, as you mentioned, gear selectivity produces censored samples when young individuals does not appear on data. This produces skewed width distributions, impacting in vBGF estimates. For this reason, we preserves the curve assuming availability data with missed size classes at 0' and '1' ages.

L154: Specify the length type, is this carapace width?

R: We specify the length type in all parts of manuscript, is carapace length.

Results:

The tables are very informative and fi gures are clear and provide all the necessary information to support the conclusions.

R: Thanks for your comment.

Fig 2 & 4: Change x-axis label to appropriate length type.

R: Done as suggested.

Fig 3 & 5: Add y-axis label to top fi gure, change bottom figure y-axis label to 'carapace width (mm)', and label them appropriately as a and b.

R: Done as suggested. However, we add 'carapace length (mm)' as y-axis label in the three panels.

Discussion:

1. L229-232: If females devote more energy to reproduction, wouldn't that leave less for somatic growth and so they would grow slower? I think what you're referring to is an evolutionary strategy. Egg production is limited by body size and so it's an advantage for females grow faster and reach sexually maturity/capacity, and also to improve survival and competition. But they reach smaller sizes because of energy allocation. Males don't have to worry about egg capacity, so they grow slower but reach larger sizes. Consider rephrasing these sentences.

R: Thanks for your suggestion. We rephrase these sentences according your observation at lines 236-238. Also, please consider the new lines 238-241 of this discussion.